# Yggdrasil: Bridging Dynamic Speculation and Static Runtime for Latency-Optimal Tree-Based LLM Decoding

**Yue Guan**[1,2,*] **Changming Yu**[1,2,*]**, Shihan Fang**[1]**, Weiming Hu**[1,2]**, Zaifeng Pan**[3]**,**
**Zheng Wang**[3]**, Zihan Liu**[1,2]**, Yangjie Zhou**[1,2]**, Yufei Ding**[3]**, Minyi Guo**[1,2]**, Jingwen Leng**[1,2]

[1]Shanghai Jiao Tong University
[2]Shanghai Qizhi Institute
[3]University of California, San Diego
{bonboru,fichmi,fang-account,weiminghu,
altair.liu,yj_zhou,guo-my,leng-jw}@sjtu.edu.cn,
{zapan, zhw100, yufeiding}@ucsd.edu

## Abstract

Speculative decoding improves LLM inference by generating and verifying multiple tokens in parallel, but existing systems suffer from suboptimal performance due to a mismatch between dynamic speculation and static runtime assumptions. We present Yggdrasil, a co-designed system that enables latency-optimal speculative decoding through context-aware tree drafting and compiler-friendly execution. Yggdrasil introduces an equal-growth tree structure for static graph compatibility, a latency-aware optimization objective for draft selection, and stage-based scheduling to reduce overhead. Yggdrasil supports unmodified LLMs and achieves up to $3.98\times$ speedup over state-of-the-art baselines across multiple hardware setups.

## 1 Introduction

Generative large language models (LLMs)[3] have demonstrated remarkable performance across a wide range of domains, including language understanding[10], reasoning[42], and multi-modal tasks [11]. However, the increasing scale of LLMs introduces substantial computational and memory demands[52], resulting in high latency and deployment costs. To address these bottlenecks, numerous optimization strategies have been proposed[53, 28, 19]. Among them, speculative decoding[22] stands out for offering lossless acceleration by exploiting parallelism during auto-regressive generation.

Traditional decoding proceeds one token at a time, where each new token is causally dependent on the previous one. This strictly sequential nature severely underutilized modern hardware. Speculative decoding breaks this bottleneck by generating multiple candidate tokens and verifying them in parallel using the original model. If the draft matches the oracle, multiple tokens can be accepted in a single generation step, improving the overall throughput. This concept echoes classical architectural optimizations like branch prediction[38] and cache prefetching[5].

Yet despite its promise, existing speculative decoding systems fall short of optimal performance due to **a fundamental mismatch between dynamic drafting algorithms and the static assumptions of modern compiler-based runtime**. As illustrated in Fig. 1, speculative decoding dynamically adjusts tree structures and operator shapes per decoding step to maximize token acceptance. This behavior inherently conflicts with the static dataflow graphs expected by deep learning compilers, which rely on fixed computation patterns for graph fusion, kernel tuning, and memory planning.

Moreover, current methods often optimize average accepted length (AAL) under simplistic assumption which ignoring non-uniform verification costs. Meanwhile, runtime systems typically treat speculative

---

*Both authors contribute equally to this work.

39th Conference on Neural Information Processing Systems (NeurIPS 2025).

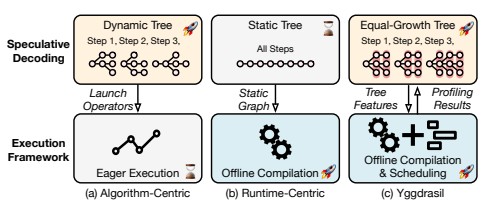

Figure 1: Overview of Yggdrasil.

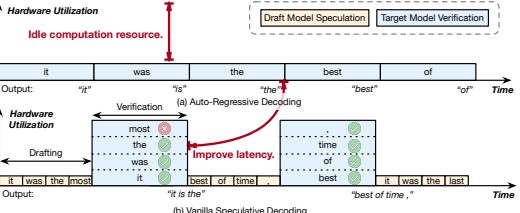

Figure 2: Illustration of speculative decoding.

decoding as a black box without addressing the nontrivial CPU logic overhead, missing opportunities to exploit speculative decoding-specific optimizations. These interlocked challenges constrain both algorithmic efficiency and system-level performance.

To overcome these limitations, we introduce Yggdrasil, a latency-optimal speculative decoding system that co-designs algorithm and runtime execution. Yggdrasil is grounded on three core ideas: (1) Latency-aware objective: a hardware-profiled optimization target that selects the optimal tree shape (depth, width, verification scope) to minimize real-world latency rather than proxy metrics like AAL (§. 4.1). (2) Equal-Growth Tree (EGT): a novel draft tree structure that ensures static operator shapes for compatibility with compile-time graph optimization, while maintaining flexibility to adapt to decoding context (§. 4.2). (3) Stage-based scheduling: a speculative-decoding-specific search framework that minimizes CPU-GPU coordination costs by eliminating conditional branches and overlapping dependent computations (§. 5).

Yggdrasil system is built upon an open-sourced deep learning compiler TorchInductor[2] requires no modification to model architectures, supporting a wide range of LLMs. Across diverse hardware and model configurations, it achieves up to $3.98\times$ speedup over state-of-the-art systems including SpecInfer, Sequoia, and vLLM-Spec. This research makes the following key contributions:

• We design an equal-growth tree drafting algorithm optimizing a latency-aware objective that enables context-aware speculative decoding while preserving graph compilation compatibility (§. 4).

• We introduce a stage-scheduled runtime execution framework that reduces overhead by rearranging and scheduling speculative decoding stages (§. 5).

• We implement and evaluate Yggdrasil on real-world models and datasets, demonstrating consistent gains over existing baselines (§. 6 & §. 7).

## 2  Background

**Drafting and Verification.**  In the speculative decoding process, as shown with Fig. 2, the *drafting* phase involves generating candidate sequences of tokens for future steps. These sequences are then subjected to the *verification* phase, where their correctness is assessed by the oracle model. The corresponding models are referred to as *drafter* and *verifier/target model*. By enabling parallel token validation, speculative decoding reduces the number of decoding iterations.

**Average Accepted Length.**  A key performance metric in speculative decoding is the *average accepted length* (AAL), which measures the average number of tokens accepted per decoding iteration. A higher AAL indicates greater efficiency, as more tokens are appended to the output.

**Source of Improvements.**  The core insight behind the effectiveness of speculative decoding lies in its utilization of underused computational resources. Many modern computing devices, such as the H100 GPU card [33], are memory-bound during decoding inference, underutilizing their computational capacity. This underutilization creates an opportunity for speculative decoding to maximize resource efficiency. Additionally, the Transformer [43] decoder architecture, which forms the backbone of most LLMs, inherently supports parallelism, making it well-suited for speculative decoding. Specifically, the self-attention mechanism within the Transformer uses a sequence-level mask to encode causal dependencies in the input sequence [43]. This design verifies the speculative sequence in a single, parallel inference execution, as illustrated in Fig. 2. The combination of speculative drafting and efficient sequence-level parallel verification enhances the decoding process.

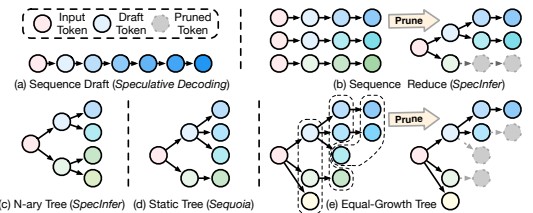

Figure 3: Comparison of drafting structures.

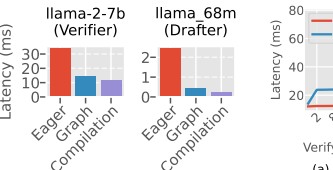

Figure 4: Benchmark of different runtimes.

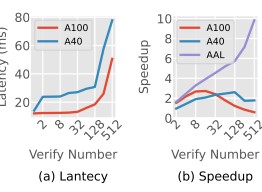

Figure 5: Latency and speedup characteristics.

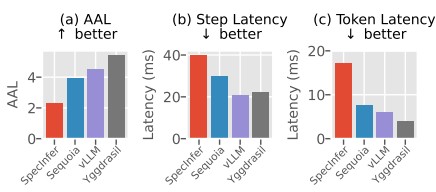

Figure 6: Comparison of AAL, per-step latency, and per-token latency.

Table 1: Comparison of prior arts.

| Research | Draft Method | | Compilation | |
|---|---|---|---|---|
| | Adaptivity | Structure | Draft | Verify |
| Speculative Decoding[22] | Static | Sequence | ○ | ○ |
| DISCO[29] | Dynamic | Sequence | ○ | ○ |
| SpecInfer[31] | Static | Tree | ○ | ○ |
| vLLM-Spec[27] | Static | Sequence | ● | ● |
| Sequoia[8] | Static | Tree | ● | ○ |
| **Yggdrasil** | Dynamic | Tree | ● | ● |

**Tree-Based Drafting.** While speculative decoding utilizes the redundant computational resources in the verification process, its generative drafting phase is still under-utilized as shown in Fig. 2-(b). One advanced approach in speculative decoding is the use of *tree-based drafting*, where predictions are organized hierarchically in a tree structure. This structure enables parallel verification and expands the exploration space for candidate tokens. While tree-based drafting significantly broadens the speculative exploration space and increases the AAL metric, its effectiveness is heavily influenced by the tree's structure and construction strategy, which impact both system performance and computational overhead.

SpecInfer [31] proposes generating multiple draft sequences with individual drafter models and reducing them into a tree structure. While this approach expands the exploration space, it introduces duplicate speculations, resulting in increased drafting overhead. An alternative approach from SpecInfer involves constructing a K-ary tree by sampling the top-K tokens from the prediction distribution at each drafting step. This method covers a wide range of speculative possibilities and guarantees high AAL. However, it becomes impractical for deeper trees due to their exponential complexity, which limits their draft length.

DISCO[41] introduces a fully dynamic tree structure, where the tree's depth and width are determined on-the-fly based on sequence context. This dynamic approach achieves high AAL under a fixed verification budget but incurs significant runtime overhead due to its reliance on dynamic control flows and variable operation shapes, as further discussed in §. 3.

Sequoia [8] takes a different approach by proposing a dataset-adaptive tree structure that balances AAL and runtime overhead. By profiling the optimal tree structure for a given verification budget and dataset distribution, Sequoia optimizes performance. However, it applies the same tree structure to all input tokens, disregarding the variability in drafting exploration space across different contexts.

## 3 Motivation

Although speculative decoding has a significant potential to accelerate LLM inference, existing systems fail to achieve optimal performance due to a fundamental gap between algorithm design and runtime support. To elaborate, we highlight two bottlenecks in the following.

**Dynamic drafts break static compilation.** As explained in §. 2, context-aware tree drafting boosts AAL without increasing the number of verification tokens. Yet this dynamism collides head-on with the static-graph assumptions baked into today's compiler optimizations, as illustrated in Fig. 4. For instance, capturing Llama-2-7B with CUDA Graphs delivers a $2.32\times$ speedup, but the control-flow branches introduced by dynamic drafting cannot be frozen into a single graph, blocking this gain. Operator-level auto-tuning[17] suffers in the same way. Kernels compiled for one fixed shape yield an extra $1.23\times$ speedup, but only when shapes stay constant. In short, the very flexibility that raises AAL deprives us of the compile-time optimizations that make LLM inference fast.

**High AAL is not equal to high end-to-end speedup.** Most prior work maximizes AAL under the tacit assumption that more accepted tokens translate linearly into wall-clock savings. That equivalence holds only when verification remains cheap, as in Fig. 5-(a). Once the number of verification tokens grows, the verification latency per step escalates. In this case, we can still observe AAL speedup while the actual per-token speedup curve flattens and eventually reverses(Fig. 5-(b)). Optimizing AAL in isolation produces diminishing or even negative returns. Any realistic objective must weigh acceptance gains against verification overhead in the current decoding context.

**Summary.** These bottlenecks translate into a sub-optimal speculative system design as shown in Fig. 6. Tree-based approaches such as Sequoia push AAL high but retain high per-step latency. Compilation-oriented systems like vLLM [27] and TRT-LLM [34] slash step latency through static kernels yet can't exploit dynamic speculation, capping AAL. These all lead to a sub-optimal per-token latency and no existing framework simultaneously delivers both high AAL and low runtime latency, exposing an urgent need for techniques that reconcile dynamic drafting with compilation efficiency.

## 4 Latency-Optimal Tree Speculation

In this section, we introduce a novel tree-drafting method to address the challenges of dynamic speculative decoding. We first present a latency-aware optimization objective for speculative decoding, which maximizes the actual system wall time speedup. Then, we introduce the context-aware dynamic tree drafting algorithm that is friendly to compile-time optimizations.

### 4.1 Latency-aware Optimization Objective

As discussed in §. 3, existing systems usually maximize the AAL as an approximation for the system speedup.

$$Speedup_{\text{naive}} = \frac{T_{\text{generative}}}{T_{\text{speculative}}} \simeq \frac{Num_{\text{iteration}} * T_{\text{target}}}{T_{\text{target}}} = AAL \tag{1}$$

where $Num_{iteration}$ is the number of forward inference steps executed in the naive generative paradigm to achieve the equal output length of speculative decoding, which is equal to the AAL. However, this approximation overlooks (1) the overhead of drafting iterations and (2) the latency characteristics of the models.

As such, we propose adopting a more accurate speedup metric that considers the actual execution latency. We explain it begin with the vanilla speculative decoding setting as following.

$$Speedup = \frac{T_{\text{generative}}}{T_{\text{speculative}}} = \frac{AAL * T_{\text{verifier}}(1)}{Num_{\text{draft}} * T_{\text{drafter}} + T_{\text{verifier}}(Num_{\text{draft}} + 1)} \tag{2}$$

The $T_{\text{verifier}}(W)$ is the verification time of the verification model with $W$ tokens verified in parallel as shown in Fig. 5-(a). And the $Num_{draft}$ is the number of inference iterations of the drafter model to produce the speculation, which is greater or equal to AAL-1. The -1 term here originates from the bonus token of the verification model. However, this term is non-trivial to estimate as it is dependent on the hardware and model size. Previous works simplify it by strictly restricting the verification number to be smaller than a threshold and treat it as a constant. This simplification overlooks the opportunity to generate more tokens than the threshold with the extra verification time, which is beneficial when the current speculation is of high quality.

The problem is further complicated with the tree speculation since the AAL and $T_{\text{draft}}(W_{\text{draft}})$ is dependent of the tree structure, specified as follows.

$$Speedup = \frac{AAL(W_{\text{draft}}, D_{\text{draft}}, W_{\text{verify}}) * T_{\text{verifyer}}(1)}{\sum_{D_{\text{draft}}} T_{\text{drafer}}(W_{\text{draft}}) + T_{\text{verifier}}(W_{\text{verify}})} \tag{3}$$

where $W_{\text{draft}}, D_{\text{draft}}, W_{\text{verify}}$ are tree-related settings representing the width and depth of the tree. In specific, the depth of the tree determines the number of drafting iterations to launch. And the width of the tree determines the number of tokens to be drafted in parallel. With this objective, we get a better estimation of the system latency that reflects the actual dynamic tree structure.

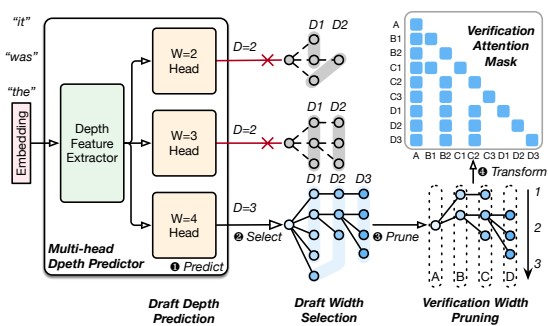

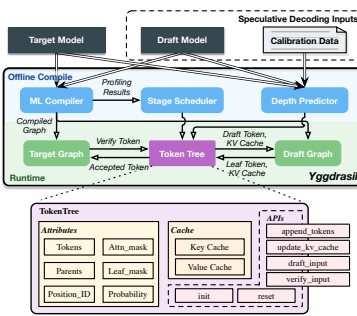

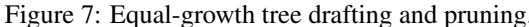

Figure 7: Equal-growth tree drafting and pruning.    Figure 8: System implementation.

## 4.2   Equal-Growth Tree

The challenge is to find the optimum tree structure that maximizes the speedup objective by adapting to the context while being friendly to compile-time optimizations. In the previous section, Equation 3 reveals that speedup depends not only on the AAL, but on the tightly coupled trio $\langle W_{\text{draft}}, D_{\text{draft}}, W_{\text{verify}} \rangle$. Searching this high-dimensional space exactly is intractable, so we break the problem into three greedy—but coordinated—sub-decisions and introduce the Equal-Growth Tree (EGT) algorithm to bind them together.

EGT predicts the $D_{\text{draft}}$ and launches all draft graphs to eliminate per-token control-flow stalls. We then select the $W_{\text{draft}}$ that maximizes the speedup objective under the predicted tree depth. Finally, we prune the draft tree to derive a subtree with size $W_{\text{verify}}$ that maximizes the speedup objective. Together these steps yield a *context-adaptive yet runtime-friendly* tree that outperforms prior dynamic-sequence methods[30, 27] in both AAL and throughput.

**Draft Depth Prediction.** We first predict $D_{\text{draft}}$, i.e. the number of draft model invocations. A lightweight multi-head depth predictor is inserted with a two-layer MLP encoder and multiple depth prediction heads. The predictor consumes the target model's last-token embedding and outputs the expected acceptance length. We train this predictor offline for each dataset and drafter/verifier pair with training data collected once via profiling on an in-domain validation corpus. Unlike iteration-wise approaches, we instantiate all $D_{\text{draft}}$ draft graphs concurrently, eradicating CPU branch mispredictions and maximizing GPU overlap.

**Draft Width Selection.** We then select $W_{\text{draft}}$ to maximize the speedup objective under the predicted tree depth. This is a greedy selection balance the trade-off between the AAL and the verification time by selecting the optimal width with predicted depth. While EGT grows exactly $W_{\text{draft}}$ leaves per draft step, we may attach them *anywhere* in the partial tree: we choose the positions whose path-wise expected AAL gain is largest, using generation probabilities as surrogates for acceptance likelihood[44].

**Verification Width Pruning.** After drafting, we solve a maximum-value subtree problem to respect the verification number $W_{\text{verify}}$. Since the other terms in Eq.3 are determined at this point, the problem is reduced to finding a sub-tree in the drafted speculation tree that maximizes the overall speedup objective. This is solved with a dynamic programming algorithm that traverses the tree in a bottom-up fashion, computing the speedup value for each node and its children. This post-processing is necessary only because online equal-growth is greedy predicted, because if the oracle optimum were known during growth, pruning would be unnecessary.

Lastly, we prepare the generated tree structure for parallel verification by generating the attention mask adhering to the tree's causal dependency[36].

## 5   Stage-based Scheduling Runtime

With the EGT, we can get a runtime-friendly drafting algorithm that launches static computation graphs for GPU execution. However, this is not the end of the story. The complex execution flow and dynamic control logic of speculative decoding causes heavy CPU overhead as shown in Fig. 9-(a). That means some GPU operation are launched dynamically by the CPU evaluation results, which lead to bubbles on the GPU wall-time. For example, based on the verification results, we apply

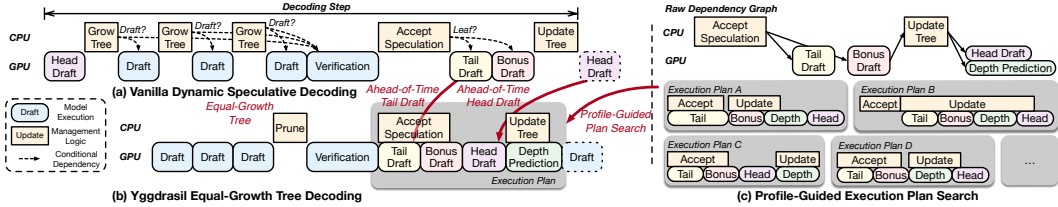

Figure 9: Stage-based speculative decoding scheduling runtime.

the accept speculation management on CPU and determine whether to draft for the last tail token on-the-fly. This leaves the GPU idle when applying the accept speculation stage. To this end, we discuss two speculative decoding-specific runtime optimizations that originates from the dynamic and multi-stage nature of the speculative decoding. We first breaks the dependency between some stages by speculation to bring its execution ahead for better overlapping efficiency. And then we search for the best-performance execution plan under current EGT setting with profiling statistics.

## 5.1 Ahead-of-Time Stage Execution

In the naive speculative-decoding pipeline pronounced idle periods arise from control-flow dependencies, as shown in Fig. 9-(a). Our EGT approach mitigates most bubbles in the drafting phase, but the post-verification validation step still suffers heavy CPU overhead. We therefore overlap CPU management with GPU inference to reduce wall-clock time. The challenge is that speculative decoding contains data dependencies that seemingly preclude such overlap—for example, we must know the set of accepted tokens before identifying the next head token to draft, partitioning execution into discrete stages (Fig. 9-(c)). Our key observation is that several of these dependencies can be speculatively broken by executing downstream stages with a superset of the tokens actually required. Concretely, we advance two stages so they can proceed ahead-of-time.

**Ahead-of-Time Tail Draft.** Instead of conditionally drafting only the final tail token, we speculatively draft the entire candidate sequence. Once any leaf is accepted, we simply reuse the corresponding results, eliminating the conditional branch. Because both acceptance and tail drafting are lightweight, they can run concurrently. This removes the tail-draft CPU logic from the critical path.

**Ahead-of-Time Head Draft.** Head drafting normally decodes a single token using the confirmed root, introducing a small GPU kernel on the execution critical path. We instead propose drafting on the entire forthcoming sequence, issuing the head draft immediately after the previous iteration's bonus draft. The released dependency lets us overlap head drafting with the acceptance stage, further compressing the execution timeline.

## 5.2 Profile-Guided Execution Plan Search

While ahead-of-time execution unlocks overlap, it also inflates model execution overhead by processing extra tokens. The trade-off depends on the EGT parameters, the relative cost of each stage, and the target hardware. This results in a large and highly environment-dependent design space. We tackle this with an offline, profile-guided search framework that explores stage-overlap strategies within the block region of Fig. 9-(b). Upon the raw dependency graph shown in Fig. 9-(c), we have the ahead-of-time executions breaking the dependency and the parallel stages that are free to schedule. As such, we can search for the best execution plan with profiling results of each stage that estimates the end-to-end latency of each candidate schedule. For the ahead-of-time stages, moving a stage forward introduces more drafting computation but creates additional overlap opportunities, making the sweet spot non-trivial. We keep the profiling metrics of both the original and ahead-of-time execution stages and use a simple grid search to find the best execution plan. Thanks to the well-defined dependency graph, the search space is small and can be done offline at compile time. We visualize several representative plans in Fig. 9-(c).

## 6 Implementation

Fig. 8 showcases the architecture of Yggdrasil, which is meticulously designed to optimize speculative decoding for modern inference systems. The system is divided into two tightly integrated

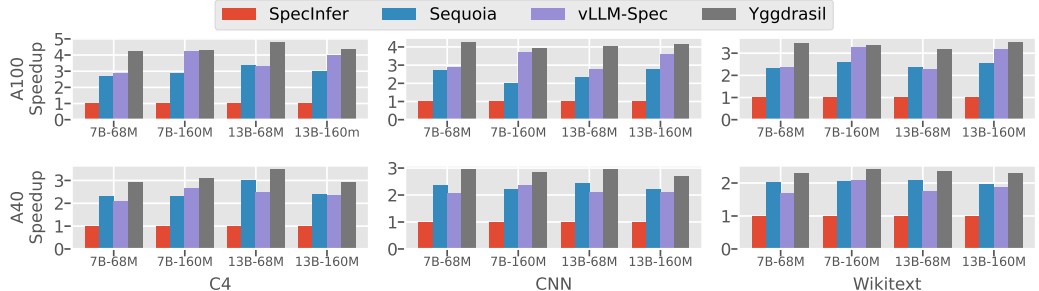

Figure 10: End-to-end per-token generation latency speedup results over SpecInfer[31].

subsystems: (1) compile-time optimization modules and (2) runtime execution modules. At compile time, users provide a draft model, a verifier model, and a small calibration dataset. From this point forward, the workflow operates in a fully model-agnostic manner, requiring no source-level modifications to the original networks. Yggdrasil leverages state-of-the-art ML compilers to lower both models and employs profile-driven cost models to generate an execution schedule optimized for the target hardware. Additionally, it trains a lightweight depth predictor to guide speculative decoding, ensuring efficient and accurate execution.

At runtime, the core abstraction is the `TokenTree` class, which encapsulates the semantics of the EGT. The `TokenTree` seamlessly interacts with the draft and verifier models to generate speculative tokens and verify their correctness. It provides robust interfaces for exchanging token sequences and their corresponding KV cache tensors with the draft and verifier models. The KV cache storage, along with other critical metadata such as the tree structure array, attention mask, and leaf positions, is efficiently managed as properties of the `TokenTree`.

The Yggdrasil system is implemented on the PyTorch[2] framework, utilizing the TorchInductor compiler backend to achieve high performance. Importantly, the proposed dynamic speculation tree generation algorithm and pipeline runtime are designed to generalize across a wide range of inference systems[34, 20], making Yggdrasil a versatile and impactful solution for modern AI workloads.

## 7 Evaluation

### 7.1 Experimental Setup

**Testbed.** The evaluation of Yggdrasil encompasses servers featuring GPU cards of NVIDIA A100 (80G), A40 and Intel(R) Xeon(R) E5-2620 v3 @ 2.40GHz CPU. Our experiments rely on essential dependencies: CUDA-11.7 and TorchInductor.

**Benchmark and Datasets.** The proposed Yggdrasil system is designed to work seamlessly with various target models and datasets, ensuring broad applicability. While factors like model alignment and task complexity influence AAL and speedup, these are orthogonal to the system design. Yggdrasil consistently outperforms benchmarks across diverse settings. To demonstrate its effectiveness, we evaluate it on language modeling datasets such as C4 [39], Wikipedia [45], and CNNDaily [18]. We use Llama-2-7B and Llama-2-13B as target models, paired with drafting models Llama-68M and Llama-160M [31], focusing on per-token latency (TPOT) as the primary metric.

**Baselines.** We evaluate the proposed Yggdrasil system by comparing it with a wide spectrum of related systems. For the end-to-end system performance, we compare it to state-of-the-art speculative decoding systems and LLM serving runtimes, including vLLM [20] and FlexFlow [31]. To benchmark the effectiveness of the proposed tree structure, we conduct experiments with reference to tree structures from the algorithm communities, including Sequoia [8] and SpecInfer [31].

### 7.2 LLM Decoding Latency Benchmark

We evaluate the end-to-end latency performance of Yggdrasil for LLM decoding, focusing on per-token latency to assess decoding efficiency. The results, shown in Fig. 10, highlight that Yggdrasil consistently outperforms all baselines, achieving average decoding latency improvements of $3.98\times$

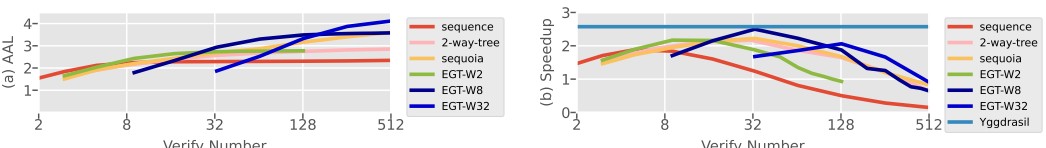

Figure 11: AAL and speedup comparison of tree structures on Wikitest dataset.

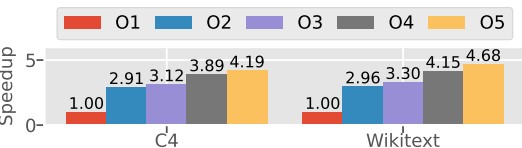

Figure 12: Optimization breakdown on Llama-2-7B model with Llama-68m as the drafter.

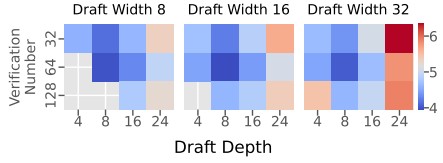

Figure 13: EGT parameter sensitivity analysis with Llama-2-7B and Llama-68m.

on A100 GPUs and $2.76\times$ on A40 GPUs. This demonstrates the effectiveness of its co-designed approach, surpassing both algorithm-centric designs like Sequoia and system-centric SpecInfer.

Among the baselines, SpecInfer performs the worst due to significant overhead from GPU execution and serving runtime. Sequoia and vLLM-Spec leverage TorchInductor [2] for optimized model execution, achieving average speedups of $2.45\times$ and $2.66\times$ over SpecInfer, respectively. However, Yggdrasil further improves the speedup to $3.37\times$ by incorporating context-aware speculation and CUDA-graph optimizations. Notably, the performance gains are more pronounced on A100 GPUs, which suffer from greater hardware under-utilization, allowing Yggdrasil to maximize its benefits.

## 7.3 Tree Structure Comparison

We then evaluate the efficacy of the optimization techniques proposed in this work, delivering the end-to-end performance improvement together. We first benchmark the efficacy of the proposed EGT tree structure and the hardware-aware optimization objective.

**Tree Structure.** We evaluate the AAL of various tree structures in Fig. 11-(a). The sequence speculative decoding paradigm shows limited AAL and quickly saturates as the verification number increases, restricting its speedup potential compared to tree-based structures. Sequoia, the SOTA baseline, achieves good AAL by statically adapting its tree structure based on the dataset and verification number. However, it uses the same tree structure for all inputs, ignoring context. In contrast, EGT dynamically adjusts its width, consistently outperforming Sequoia's static tree under optimal verification settings, highlighting the importance of context-aware tree structure selection.

**Context-aware Dynamic Tree.** We evaluate the impact of Yggdrasil's dynamic tree structure selection process using the theoretical speedup from Eq.3 to approximate tree structure effectiveness. The results in Fig. 11-(b) show that sequence speculative decoding achieves the lowest speedup, highlighting the need for tree structures. Static trees like N-way and Sequoia degrade as verification numbers increase due to higher verification latency from resource saturation (§. 6). In contrast, EGTs with varying widths achieve optimal speedup under different verification settings. By dynamically selecting the best width and verification number for each request, Yggdrasil consistently outperforms all baselines, leveraging context and hardware characteristics for superior performance.

## 7.4 Optimization Breakdown

We then analyze the effectiveness of the proposed optimization techniques in the Yggdrasil system. We break down the overall performance improvement into five optimizations as shown in Fig. 12. Their performance and synergy are discussed in the following.

**O1** is only applying the **latency-optimal tree speculation** introduced in §. 4. As its improvement largely correlates to the runtime support, we benchmarked their effectiveness independently with other tree structures in the previous subsection. Here, O1 could be considered a general baseline, with its improvement reflected by the AAL metrics.

**O2** adopts **graph compilation** with the TorchInductor compiler for the drafter and verifier. We find that O2 accounts for an average of $2.775\times$ speedup, which is the most significant of the four

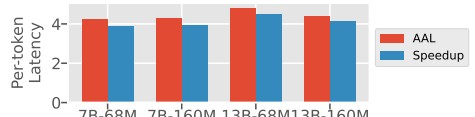
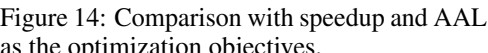

Figure 14: Comparison with speedup and AAL as the optimization objectives.

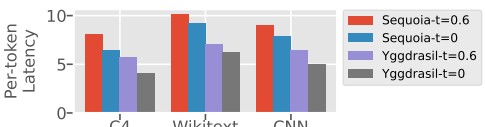

Figure 15: Latency with different sampling temperatures using Llama-7B and Llama-68M.

optimizations. This stresses the necessity of runtime support for the speculative decoding system. Note that the graph can only be achieved with the EGT design to deliver high AAL simultaneously. Compared to the end-to-end results, we find that using a trivial yet runtime-friendly sequence structure to fully utilize the compilation optimization outperforms some complex speculation algorithms. Yggdrasil bridges the optimizations to embrace improvements from both.

**O3** applies **verification width pruning** using the speedup objective to enhance context flexibility. By dynamically adjusting the verification number, this optimization achieves an average speedup of $1.07\times$ over O2. The improvement primarily stems from the adaptive verification strategy, which reduces overhead by aligning the verification number with the saturation region of the inference wall time curve (Fig. 5). This highlights the effectiveness of adaptive verification, a technique applicable to all speculative decoding systems.

**O4** accounts for the **graph-based scheduling** optimization discussed in §. 5. It achieves an average of $1.21\times$ improvement compared to the previous optimizations. Note that this is measured on the per-token equivalent latency generated by one decoding iteration. Since the number of accepted tokens scales with the wall time of all stages in the iteration, this optimization is significant to the system's performance. This suggests the importance of exploiting the speculative-decoding specific optimizations within the existing execution runtime.

**O5** is the **draft depth predictor** that predicts the optimal drafting steps given the contextual embedding. To compare, we use a fixed tree structure of depths of 16 and width of 8 as a baseline when O5 is excluded. We will further analyze the impact of the tree settings below. We find that the predictor brings a $1.10\times$ speedup to the system by explicitly considering the difficulty of the context.

### 7.5 Sensitivity Analysis

**EGT Structure.** We conduct a sensitivity analysis of the EGT parameters, as shown in Fig. 13. In §. 4.2, we described our method for determining optimal parameters at runtime. Here, we provide an intuitive exploration of the parameter search space, excluding invalid configurations. The results reveal that the interplay between draft width, draft depth, and verification number significantly impacts the achieved per-token latency. For instance, the combination $D_{draft} = 8$, $W_{draft} = 8$, and $W_{verify} = 64$ yields the best performance in this static analysis.

**Speedup Objective.** Similarly, we conduct an ablation experiment to show the advantage of optimizing the speedup objective rather than AAL on the C4 dataset shown in Fig. 14. Incorporating Eq. 3 demonstrates an extra 8% performance improvement compared to optimizing AAL directly across all drafter-verifier settings. This is because compared to AAL, taking the speedup as the optimization objective enables better capturing of the runtime characteristic, including the dynamic draft and verifier execution time.

**Temperature Impact.** We study the impact of adjusting the sampling temperature values on Sequoia and Yggdrasil. As Figure 15 shows, both Sequoia and Yggdrasil obtain better performance when the temperature is 0. The reason is that when the temperature is low, the draft model can align better with the target model, resulting in a higher AAL. Besides, across different temperatures, Yggdrasil consistently outperforms Sequoia, achieving an average speedup of $1.49\times$.

## 8 Related Work

**Speculative Decoding Variants and Serving Systems.** Besides the work discussed in §. 3, there is a broad range of research proposing variants of speculative decoding. First, *model-transparent* researches improve the draft accuracy or efficiency without modifying the structure of the target model. Lookahead decoding[14], ANPD[35], and NEST[23] apply more efficient retrieval or statistical

methods to produce draft tokens instead of the draft model. Several works[31, 41] design more complicated sampling procedures for better drat accuracy. These works are orthogonal to Yggdrasil. In contrast, *model-invasive* methods modify the target model to either align the target and drat models [51] or produce the speculations using parts of the target model [26, 12] or extra submodules [4, 25] to reduce draft overhead. These works require explicit modification of the origin model and are beyond the supporting scope of this work. Additionally, aside the serving systems [31, 8, 20] benchmarked in §. 7, SmartSpec [27] and MagicDec [6] study combining speculative decoding and continuous batching and achieving balance between the service level objective (SLO) and throughput improvement in online serving scenarios. However, they do not exploit the execution efficiency of the speculative decoding itself, which is the main focus of this paper.

**Dynamic Support in ML Systems.** Many works study handling the dynamic workloads in machine learning systems. Frameworks like TensorFlow Eager[1] and PyTorch Eager[37] enable dynamic graph construction[16] at runtime but suffer from suboptimal hardware utilization. ML compilers, including PyTorch 2.0[2], DietCode[49], Nimble[21], and BladeDISC[50] optimize the models with varying input shapes with adaptive graph transformation and code generation. For models with data-dependent execution paths, Cocktailer[47], Grape[48], Cortex[13], BrainStorm[24] and DyCL[7] optimize dynamic control flows to reduce the runtime overhead. Additionally, in dynamic serving scenarios, DyNet[32], BatchMaker[15], DVABatch[9], Brainstorm[24], ORCA[46], and Llumnix[40] improve GPU utilization and throughput by dynamically adjusting batch sizes or optimizing scheduling. Despite the above advancements, these works still face limitations when handling the additional overhead of speculative decoding, where unpredictable batch sizes and control flows add significant complexity.

## 9    Limitations

The core results of Yggdrasil are derived under a latency-optimal setting where a single interactive request monopolizes all available GPU memory and compute. This configuration mirrors edge or on-prem deployments where a dedicated accelerator serves one user or a latency-critical pipeline. While this assumption enables a clean analysis of memory-bandwidth trade-offs and motivates the EGT drafting, it is not applicable for the batched, throughput-oriented serving. Production inference stacks typically batch tens to hundreds of prompts to maximize accelerator utilization. In that regime, co-coordinating speculative decoding with batch schedulers becomes essential to overall efficiency. Designing a unified policy that jointly decides when to speculate and how to pack requests is paramount. In short, while Yggdrasil pushes the latency frontier for single-request scenarios, extending the framework to the latency-throughput joint optimization space, where speculative decoding and batch scheduling are solved together, remains an open line of research.

## 10    Conclusion

In this work, we present Yggdrasil, a novel latency-optimal speculative decoding framework that enables context-aware speculation while preserving compile-time optimizations. Additionally, it dynamically minimizes execution latency by adaptively selecting the optimal draft tree based on profiling-driven runtime insights. To further improve efficiency, Yggdrasil employs a stage-based execution scheduling strategy, streamlining the speculative decoding workflow. Experimental evaluations demonstrate that Yggdrasil achieves substantial performance improvements, delivering superior speedups over state-of-the-art systems.

## Acknowledgment

This work was supported by the National Natural Science Foundation of China (NSFC) Grants 62222210 and Shanghai Qi Zhi Institute Innovation Program SQZ202316. Any opinions, findings, and conclusions in this paper are those of the authors only and do not necessarily reflect the views of our sponsors. Jingwen Leng is the corresponding author. The authors express their gratitude to the anonymous reviewers for their insightful feedback, which greatly contributed to this work.

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
