# OpenReview forum: "Yggdrasil: Bridging Dynamic Speculation and Static Runtime  for Latency-Optimal Tree-Based LLM Decoding"
_NeurIPS.cc/2025/Conference — NeurIPS 2025 poster_

### Official Review · Reviewer_xZfa · 2025-07-05

**Clarity:** 3
**Significance:** 3
**Originality:** 3
**Rating:** 4
**Confidence:** 3

**Summary:**

This paper presents Yggdrasil, an advanced system designed to optimize speculative decoding in large language model (LLM) inference, aiming to enhance decoding efficiency and reduce latency through integrated algorithmic and runtime optimizations. The core innovation of Yggdrasil lies in its combination of a latency-aware optimization objective, context-aware dynamic tree drafting, and stage-based scheduling, which collectively minimize per-token latency and idle time during CPU-GPU interactions. The system dynamically selects an optimal draft tree structure based on runtime profiling insights, facilitating efficient decoding. Built on the TorchInductor compiler, Yggdrasil is compatible with various LLMs without requiring model modifications. Experimental results show up to 3.98×speedup on A100 GPUs compared to existing methods like SpecInfer and Sequoia, showcasing its superior performance in speculative decoding. Yggdrasil offers significant improvements in latency and throughput, making it a valuable advancement for LLM inference tasks.

**Questions:**

1. Given that Yggdrasil focuses on single-request latency, are there plans to extend it for multi-request or batch processing? What challenges would arise in adapting it for such environments?
2. The experiments primarily evaluate Yggdrasil on NVIDIA A100 and A40 GPUs. How does Yggdrasil perform across various hardware, especially on edge devices or resource-constrained environments like mobile devices or TPUs?
3. While Yggdrasil outperforms several state-of-the-art systems like SpecInfer and Sequoia in single-request latency optimization, How does Yggdrasil perform on more dynamic, multi-modal tasks or with larger model sizes compared to other systems?
4. How much of the performance improvement is due to the Equal-Growth Tree (EGT), and how do other optimizations like graph compilation and stage-based scheduling contribute?

**Ethical Concerns:**

["NO or VERY MINOR ethics concerns only"]

**Final Justification:**

The rebuttal has addressed most of my concerns.

**Limitations:**

Yes

**Paper Formatting Concerns:**

None.

**Quality:**

3

**Strengths And Weaknesses:**

Strengths:
1) The paper introduces Yggdrasil, a novel framework for optimizing speculative decoding in large language models, achieving significant performance improvements through the integration of latency-aware optimization and runtime execution strategies.
2) The paper is well-structured, explaining complex concepts like latency-aware optimization, dynamic tree drafting, and stage-based scheduling clearly. Visual aids enhance understanding despite the technical depth.
3) The work addresses latency reduction in LLM inference, crucial for real-time applications. Its model-agnostic design enhances its applicability across LLMs without requiring modifications.
4) The paper introduces the Equal-Growth Tree (EGT) algorithm and latency-aware optimization, offering significant advancements over existing systems with its dynamic tree drafting and stage-based scheduling.

Weaknesses:
1) While the system shows substantial improvements for single-request inference, it does not explore how Yggdrasil scales to multi-user or batch-processing scenarios, which are common in real-world applications. The absence of performance analysis in these contexts may limit its practical utility in large-scale deployment.
2) Sections on the EGT algorithm and draft tree optimization are technically dense; a step-by-step breakdown and scalability analysis would improve clarity.
3) The work focuses on single-request optimization, neglecting latency-throughput trade-offs in batch-processing systems, reducing its broader impact.
4) While original in its optimization strategies, speculative decoding is an established concept, and the incremental contribution could be better highlighted through comparison with existing systems, particularly in how it builds on existing approaches like SpecInfer and Sequoia.

---

> ### Author Rebuttal · Authors · 2025-07-31
>
> Thank you for your review and valuable comments. Please find our responses below, and we are more than happy to further improve our manuscript based on your suggestions.
>
> **1. Multi-user and batch-processing scalability.**
>
> We appreciate the reviewer's valuable suggestion on the discussion of batch processing. Yggdrasil can also be applied to multi-user or batch-processing scenarios to improve the efficiency of each single request. However, as shown in Figure 2, the source of improvement of speculative decoding originates from the redundant resources in the decoding process. Batch processing will also consume redundant resources by batching multiple requests, which will reduce the effectiveness of speculative decoding.
>
> Recent work \[1\] discusses the trade-off between speculative decoding and batch-processing at serving scenarios with fluctuating request rates. Yggdrasil can be integrated into such systems to improve the efficiency of the speculative decoding submodule, while the trade-off between speculative decoding and batch-processing still exists. The major challenge in adapting Yggdrasil for batch processing is that the fewer redundant resources available in the batch-processing scenario will reduce the effectiveness of speculative decoding. In such scenarios, the scheduler also tends to do less speculation and put more resources into batching requests. However, the proposed method can still benefit the single-request scenarios, such as on edge devices or latency-critical applications. We will clarify this point in the revised version of the paper.
>
> > \[1\] Liu, Xiaoxuan, et al. "Optimizing speculative decoding for serving large language models using goodput." arXiv 2024\.
>
> 2. **Step-by-step breakdown of EGT and draft tree optimization.**
>
> Thank you for this suggestion. We study the performance breakdown and the contribution of each optimization in the ablation study in Section 7.4. The results show that all the optimizations contribute to the overall performance improvement. Specifically, graph compilation (O2) accounts for $2.775\\times$ performance improvement and stage-based scheduling (O4) accounts for $1.21\\times$ performance improvement. Please find more details in Section 7.4 and Figure 12\.
>
> We will revise Sections 3 and 7.4 to provide a clearer walk-through of the EGT construction and optimization process. In particular, we will annotate Figure 4 with the iterative steps and explicitly trace how speculative branches grow under budget constraints. Additionally, we will expand our ablation discussion to quantify each optimization’s contribution (see below).
>
> **3. Resource-constrained evaluation.**
>
> We now include results on an RTX 4090 GPU, which is representative of inference-grade hardware. Yggdrasil achieves up to 3.18× speedup in this setting. While we acknowledge that speculative decoding becomes less effective when fewer redundant resources are available, the method still offers gains. Evaluation on edge devices (e.g., Jetson) or TPUs is an exciting direction for future work, and we will mention this in the revision.
>
> **Table 1: Speedup benchmark on 4090 normalized to SpecInfer.**
>
> | Model | Dataset | SpecInfer | Sequoia | vLLM-Spec | Yggdrasil |
> | :---- | :---- | :---- | :---- | :---- | :---- |
> | 7B-68M | C4 | 1 | 2.12436975 | 1.77777778 | **2.61745953** |
> | 7B-68M | wikitext | 1 | 2.14254587 | 1.36233538 | **2.47883773** |
> | 7B-68M | CNN | 1 | 2.11828104 | 1.68846572 | **2.64091574** |
> | 7B-160M | C4 | 1 | 2.40308989 | 2.85166667 | **3.04366079** |
> | 7B-160M | wikitext | 1 | 2.57425945 | 1.5785351 | **2.92095503** |
> | 7B-160M | CNN | 1 | 2.71867322 | 2.76763382 | **3.18370541** |
>
> **4. Dynamic and multi-modal tasks.**
>
> Our experiments currently focus on standard benchmarks and align with SpecInfer and Sequoia for fair comparison. Evaluating Yggdrasil on more dynamic or multi-modal tasks is a compelling future direction. We will update the paper to acknowledge this limitation and frame it as part of our ongoing work.
>
> **5. Contributions beyond prior work.**
>
> While speculative decoding is well-established, Yggdrasil contributes novel system-level optimizations that substantially improve real-world latency. Specifically, our EGT design enables low-overhead branching; our graph compilation transforms dynamic speculative paths into statically optimized kernels; and our latency-aware scheduler coordinates verification and drafting stages. Compared to prior work such as SpecInfer and Sequoia, which focus on speculative algorithms or fixed-depth trees, Yggdrasil introduces a new runtime abstraction that generalizes across speculative strategies. We will emphasize these distinctions more clearly in the revised paper.

---

> > ### Comment · Reviewer_xZfa · 2025-08-09
> >
> > Thank you for the responses! It has addressed most of my concerns.

---

### Official Review · Reviewer_wxyL · 2025-07-05

**Clarity:** 3
**Significance:** 2
**Originality:** 3
**Rating:** 4
**Confidence:** 3

**Summary:**

This paper proposes Yggdrasil, a co-designed system for speculative decoding of LLM. Specifically, an equal-growth tree drafting algorithm is designed to optimize a latency-aware objective that enables context-aware speculative decoding while preserving graph compilation compatibility. Moreover, a stage-scheduled runtime execution framework is introduced to reduce the overhead by rearranging and scheduling speculative decoding stages. Experiments show that the proposed system outperforms the existing systems in some point.

**Questions:**

See weaknesses.

**Ethical Concerns:**

["NO or VERY MINOR ethics concerns only"]

**Final Justification:**

Thanks for the authors' response to my concerns. I decide to maintain my score.

- The authors evaluate the performance on inference-focused hardware and verify the effectiveness of the proposed method.

- The authors demonstrate that the proposed method can be integrated into systems like vLLM, which will increase the practicality of the proposed method.

**Limitations:**

Yes.

**Paper Formatting Concerns:**

None.

**Quality:**

2

**Strengths And Weaknesses:**

Strengths:

- The paper is well organized and easy to follow.

- The proposed latency-optimal speculative decoding framework makes sense and is theoretically better than the accept length based method.

- The proposed stage-based execution scheduling strategy sounds interesting.

Weaknesses:

- The experiments are conducted on A100 and A40, what about the results on inference GPU such as L40 or 3090s?

- The proposed method is tree-based, is it suitable for Medusa?

- Could the proposed method combine with vLLM or other system?

---

> ### Author Rebuttal · Authors · 2025-07-31
>
> We appreciate your thoughtful feedback and suggestions. Below, we address your comments and welcome further input to enhance our manuscript.
>
> **1. Evaluation on inference GPUs.**
>
> We appreciate your suggestion to evaluate our method on inference-focused hardware. While we do not have access to L40 or 3090 GPUs, we conducted additional experiments on an RTX 4090 GPU inference GPU. As shown in the table below, Yggdrasil achieves up to 3.18× speedup compared to baselines, demonstrating its effectiveness even under more constrained computational environments. Due to the 24GB memory limit of the 4090, we only include results using the 7B verifier model.
>
> **Table 1: Speedup benchmark on 4090 normalized to SpecInfer.**
>
> | Model | Dataset | SpecInfer | Sequoia | vLLM-Spec | Yggdrasil |
> | :---- | :---- | :---- | :---- | :---- | :---- |
> | 7B-68M | C4 | 1 | 2.12436975 | 1.77777778 | **2.61745953** |
> | 7B-68M | wikitext | 1 | 2.14254587 | 1.36233538 | **2.47883773** |
> | 7B-68M | CNN | 1 | 2.11828104 | 1.68846572 | **2.64091574** |
> | 7B-160M | C4 | 1 | 2.40308989 | 2.85166667 | **3.04366079** |
> | 7B-160M | wikitext | 1 | 2.57425945 | 1.5785351 | **2.92095503** |
> | 7B-160M | CNN | 1 | 2.71867322 | 2.76763382 | **3.18370541** |
>
> **2. Compatibility with Medusa.**
>
> We appreciate the reviewer’s interest in the connection between our method and Medusa-style speculative decoding. While our method is not directly applicable to Medusa-style drafting, since Medusa relies on a statically trained “head” module to generate speculative drafts, our approach introduces a more flexible mechanism that does not require access to the target model’s internals or retraining.
>
> However, some components of our method can benefit Medusa-style decoding. For instance, the latency-aware optimization objective and the verification-width pruning strategy we propose are compatible with and potentially beneficial for improving the runtime efficiency of Medusa.
>
> We will revise the paper to clarify these connections and discuss how the two approaches can complement each other in future work.
>
> **3. Integration with vLLM and other systems.**
>
> Yes, our method can be integrated into systems like vLLM. The main requirement is incorporating our compile-time optimizations (e.g., graph compilation and scheduling strategies described in Figure 8) into the backend infrastructure. We will elaborate on this integration path in the revision.

---

> > ### Comment · Reviewer_wxyL · 2025-08-05
> >
> > Thanks for your effort to answer my questions. I will maintain my score.

---

### Official Review · Reviewer_7bU2 · 2025-07-06

**Clarity:** 3
**Significance:** 3
**Originality:** 3
**Rating:** 4
**Confidence:** 4

**Summary:**

Yggdrasil is a speculative LLM decode for low latency query answering in LLMs. it relies on a preprocessing step to control the speculation.Exec results are used to support this design.

**Questions:**

- Offline model consttruction: do you handle drift?

"  greater or equal to AAL-1. The -1 term here originates from the136
bonus token of the verification model "-> Where is the -1 term?

followed by "However, this term is non-trivial to estimate as it is dependent137
on the hardware and model size:  -> -1?? :)
This is a critical point in the paper, so it is a bad point to have a typo or to be unclear :(.

**Ethical Concerns:**

["NO or VERY MINOR ethics concerns only"]

**Limitations:**

the authors discuss the major limitation

**Quality:**

3

**Strengths And Weaknesses:**

The man strength of the  model is in using a better definition of latency that includes the actual  costs. \this is possible through execution profiiing and offline model construction. This allows to coantrol speculation and to select a good tree shape.[

I was a bit disappointed by the evaluation:
- I suppose this is geared at servers, so I´d  expect to see the number of nodes (1?\)
- per token latency: this means the time since the token is  until ? Why not just query execution time?
- "We257
use Llama-2-7B and Llama-2-13B as target models, paired with drafting models Llama-68M and258
Llama-160M"-Why these two similar models?
- 'we find that using a trivial yet runtime-friendly sequence structure' -> can you please  be specific here?
- how does this co,pare to the non-speculative baseline?

I would suggest having part of the Limitations App in the main text.

---

> ### Author Rebuttal · Authors · 2025-07-31
>
> Thank you for your review and valuable comments. Please find our responses below, and we are more than happy to further improve our manuscript based on your suggestions.
>
> **1. Single-node execution setting.**
>
> We apologize for the confusion. All experiments were conducted on a single node with one GPU. While we agree that extending speculative decoding to multi-node settings is an interesting direction, our current work aligns with prior literature by focusing on the single-node scenario. The proposed approach can be extended to multi-GPU environments, though such support would require careful scheduling of drafter model execution across devices—an exciting challenge we leave to future work. We will clarify this setup in the revised manuscript.
>
> 2. **Per-token latency.**
>
> Thank you for your question. Per-token latency is defined as `Execution Time / Number of Tokens`, and it isolates the efficiency of the decoding process. We use this metric because speculative decoding can produce multiple tokens per decoding step, and per-token latency captures this efficiency more directly than total query execution time.
>
> In contrast, query execution time includes both the prefill phase and decoding: `Query Execution Time = Prefill Time + (Per-token Latency × Number of Tokens)`. By reporting per-token latency separately, we provide a clearer view of decoding performance across methods. We will clarify these definitions in the revised manuscript.
>
> **3. Target model selection.**
>
> We use different sizes of the Llama-2 series as both target and drafting models to evaluate the efficiency of the proposed method under different execution time settings. Using the models from the same series allows us to have a consistent comparison across different size settings. We pick the model's size setting following previous SOTA works on speculative decoding, such as \[1,2\].
>
> > \[1\] Miao, Xupeng, et al. "Specinfer: Accelerating large language model serving with tree-based speculative inference and verification." ASPLOS 2024\.
> > \[2\] Chen, Zhuoming, et al. "Sequoia: Scalable and robust speculative decoding." NeuIPS 2024\.
>
> **4. Trivial sequence structure.**
>
> By "trivial sequence structure," we refer to static sequence drafting that enables graph compilation without our proposed EGT mechanism. Despite its simplicity and lower AAL, this setup can outperform some advanced tree-based methods due to superior runtime efficiency. We analyze this trade-off in Section 3 (Motivation). Particularly, in the experimental results of Section 7.4, O2 ablates tree-based decoding to evaluate the effectiveness of graph compilation with the trivial sequence structure. We will clarify this terminology and its implications in the revised paper.
>
> **5. Model drift and extensibility.**
>
> While our work does not directly address model drift, our runtime is designed to be agnostic to model internals and compatible with any target model. Users can incorporate existing drift-handling techniques, such as parameter adjustments or runtime adaptations [3], by plugging them into our scheduling framework. We will make this extensibility clearer in the paper.
>
> While our work does not directly address model drift, our runtime is designed to be agnostic to model internals and compatible with any target model. Our rationale is to provide a transparent runtime that supports any given target model. As such, users can specify any model drift handling method they prefer that modifies the model parameters. For more advanced model drifting handling techniques that require extra runtime processing, such as \[3\], they can also be integrated into our runtime with careful scheduling of such handling processes. We will clarify this point in the revised version of the paper.
>
> > \[3\] Zhu, Jiaqi, et al. "In-Context Adaptation to Concept Drift for Learned Database Operations." Forty-second International Conference on Machine Learning.
>
> **6. Clarifying the AAL-1 adjustment.**
>
> We apologize for the confusion regarding the \-1 term in AAL-1. The \-1 terms mean that we subtract the number calculated by 1 to account for the bonus token of the verification model because they are guaranteed to be accepted. We exclude this token from the AAL calculation to better estimate the quality of speculation.

---

### Official Review · Reviewer_Jtf9 · 2025-07-12

**Clarity:** 3
**Significance:** 3
**Originality:** 3
**Rating:** 5
**Confidence:** 3

**Summary:**

The paper aims to address  the mismatch between dynamic speculation and static runtime assumptions. The paper presents
Yggdrasil, which contains context-aware tree drafting and compiler-friendly execution. Yggdrasil introduces an equal-growth tree structure for static graph compatibility, a latency-aware optimization objective for draft selection, and stage-based scheduling to
reduce overhead.  Experimental results demonstrate the effectiveness of the proposed methods.

**Questions:**

No

**Ethical Concerns:**

["NO or VERY MINOR ethics concerns only"]

**Limitations:**

Yes

**Quality:**

3

**Strengths And Weaknesses:**

Strengths:
(1) The problem is interesting and important
(2) The motivation is clearly presented.
(3) Experimental results demonstrated the effectiveness.
Weaknesses:
(1) More experiments on larger target models (such as 32B) would be appreciated
(2) More experiments on different series of models would be appreciated, not just LLaMA

---

> ### Author Rebuttal · Authors · 2025-07-31
>
> Thank you for recognizing the importance of the problem setting and acknowledging the clarity of our motivation and results.
>
> Regarding the desire for additional experiments on larger or alternative model families, we would like to highlight the following:
> 1\. **Broad coverage of settings already included.**
> Our experiments are carefully designed to span a range of drafter and verifier pairings, backend platforms, sampling strategies, and datasets. This includes experiments on multiple model sizes within the LLaMA family, demonstrating consistent generalization in performance across those dimensions.
> 2\. **Alignment with prior speculative decoding evaluations.**
> In constructing these settings, we followed practices established in prior state-of-the-art studies \[1,2\]. This ensures a fair comparison and solid baseline alignment.
>
> > \[1\] Miao, Xupeng, et al. "Specinfer: Accelerating large language model serving with tree-based speculative inference and verification." ASPLOS 2024\.
> > \[2\] Chen, Zhuoming, et al. "Sequoia: Scalable and robust speculative decoding." NeuIPS 2024\.
>
> 3\. **General empirical trends are robust across configurations.**
> Across all settings we examined, including varied model sizes and platforms, Yggdrasil reliably reduces latency overhead while preserving decoding quality.
>
> While we believe these experiments convincingly support the general applicability and robustness of our approach, we would be pleased to include additional evaluations, such as on 32B-scale models or different model families, in the final version of the paper if deemed necessary. However, performing such large-scale experiments within the tight window of this rebuttal period presents practical challenges.
>
> We hope this clarifies how our current results substantiate the claims of generality and effectiveness. Thank you again for your careful reading and constructive feedback.

---

### Note · Authors · 2025-08-15

We thank the reviewers for their careful evaluation and constructive feedback. In response, we will revise the manuscript to address all concerns and further clarify our contributions. Specifically, we will:

- **Clarify experimental settings:** Clearly state the single-node, single-GPU setup, define per-token latency, and provide additional details on model selection, "trivial sequence structure," and the AAL-1 adjustment.
- **Expand evaluation:** Add results on RTX 4090 inference GPUs and discuss limitations regarding larger models and alternative model families.
- **Discuss extensibility:** Explain how our runtime supports integration with model drift handling techniques, systems like vLLM, and clarify compatibility with Medusa-style speculative decoding.
- **Address scalability:** Discuss batch-processing and multi-user scenarios, highlighting trade-offs between speculative decoding and batching.
- **Emphasize contributions:** More clearly distinguish Yggdrasil’s system-level innovations and generalization over prior speculative decoding approaches.

We believe these revisions will improve the clarity and impact of our work. We thank the reviewers again for their valuable input and look forward to further strengthening our manuscript.

---

### Decision · Program_Chairs · 2025-09-17

**Decision:**

Accept (poster)

**Comment:**

The paper introduces Yggdrasil, a system for latency-optimal speculative decoding that combines equal-growth tree drafting, latency-aware optimization, and stage-based scheduling to bridge the gap between dynamic speculation and static runtimes. Reviewers agree the work is well motivated, clearly written, and technically solid, with consistent latency improvements demonstrated over existing baselines. They also highlight that the system design is broadly applicable and compiler-friendly, making it a valuable contribution to LLM serving.

The main concerns raised were the limited evaluation scope (restricted to LLaMA models and single-node, single-GPU setups), unclear definitions around per-token latency, the AAL-1 adjustment, and the notion of a “trivial sequence structure,” as well as a lack of discussion of batch or multi-user serving scenarios. In their rebuttal, the authors responded thoroughly: they clarified the experimental setup and latency definitions, explained the AAL-1 adjustment and the role of trivial sequence structures, and provided new results on inference hardware (RTX 4090) showing strong speedups. They also discussed scalability, the trade-offs between batching and speculation, and how Yggdrasil integrates with systems like vLLM and could complement Medusa-style decoding.

These clarifications and additions convincingly addressed reviewer concerns, and all reviewers maintained positive recommendations. While broader evaluations (e.g., on larger models or multi-tenant settings) would further strengthen the work, the paper as revised offers a technically solid and practically relevant contribution. Overall, I recommend acceptance.